# A Systems Biology Approach for Personalized Medicine in Refractory Epilepsy

**DOI:** 10.3390/ijms20153717

**Published:** 2019-07-30

**Authors:** Giuseppina Daniela Naimo, Maria Guarnaccia, Teresa Sprovieri, Carmine Ungaro, Francesca Luisa Conforti, Sebastiano Andò, Sebastiano Cavallaro

**Affiliations:** 1Institute for Biomedical Research and Innovation, National Research Council, Contrada Burga, Piano Lago, 87050 Mangone (CS) and Via Paolo Gaifami 18, 95126 Catania, Italy; 2Department of Pharmacy, Health and Nutritional Sciences, University of Calabria, Rende, 87036 Cosenza, Italy; 3Centro Sanitario, University of Calabria, Via Pietro Bucci, 87036 Arcavacata di Rende (CS), Italy

**Keywords:** epilepsy, pharmacogenomics, pharmacoresistance, functional genomics, GABA_A_ receptor, drug transporters

## Abstract

Epilepsy refers to a common chronic neurological disorder that affects all age groups. Unfortunately, antiepileptic drugs are ineffective in about one-third of patients. The complex interindividual variability influences the response to drug treatment rendering the therapeutic failure one of the most relevant problems in clinical practice also for increased hospitalizations and healthcare costs. Recent advances in the genetics and neurobiology of epilepsies are laying the groundwork for a new personalized medicine, focused on the reversal or avoidance of the pathophysiological effects of specific gene mutations. This could lead to a significant improvement in the efficacy and safety of treatments for epilepsy, targeting the biological mechanisms responsible for epilepsy in each individual. In this review article, we focus on the mechanism of the epilepsy pharmacoresistance and highlight the use of a systems biology approach for personalized medicine in refractory epilepsy.

## 1. Introduction

Epilepsy is a heterogeneous and often severe neurological disorder characterized by recurrent and unpredictable seizures due to paroxysmal and synchronized discharges in a group of brain cells, located in one or both hemispheres. This pathological condition can start at any age (from childhood to adulthood), with distinct etiology, symptoms and prognosis. According to the World Health Organization, epilepsy currently affects approximately 50 million people worldwide, with a global estimate of 2.4 million new diagnosed cases each year. Overall prevalence of epilepsy roughly lies in the range of 4–10 per 1000 people, which is usually higher in low- and middle-income countries. Most patients achieve optimal seizure control through the chronic administration of antiepileptic drugs either alone (monotherapy) or in combination (polytherapy). In spite of a great variety of pharmacological agents available for epilepsy treatment, many people are non-responsive or refractory to the therapy with antiepileptic drugs, making pharmacoresistance one of the most crucial clinical problems in disease management. Generally, epilepsy is considered ‘intractable’ when seizures fail to come under control with two or more anticonvulsant treatments [1]. However, a precise definition of ‘refractory epilepsy’ has not been established. Many studies report different criteria to define epilepsy as multidrug resistant, such as the number and the dose of drugs used, the specific treatment or the period of convulsions occurrence during drug administration [2,3]. These discrepancies do not allow a suitable comparison of both clinical research data and drugs therapy. To univocally clear this condition, the International League Against Epilepsy (ILAE) suggested a consensus definition of refractory epilepsy according to which drug resistant epilepsy may be defined as “failure of adequate trials of two tolerated and appropriately chosen and used ASD schedules (whether as monotherapies or in combination) to achieve sustained seizure freedom” [4]. The inability to control seizures is associated with a high risk of cognitive deficits, psychosocial consequences, shortened lifespan, bodily injury and reduced life quality of epileptic patients [5,6]. Therefore, early identification of the causes of drug resistance to AED is fundamental to avoid the chronic manifestations of this complex disease. Initially, the heterogeneity of pharmacological treatment outcome, among epileptics with the same diagnosis and the same therapy, was attributed to a plethora of physiological and environmental factors, such us age, sex, race/ethnicity, hepatic and renal function, lifestyle, and assumption of other drugs for the treatment of concomitant diseases [7]. Nevertheless, the pathogenesis underlying pharmacoresistance in epilepsy cannot be ascribed to the aforementioned factors only.

Among the different hypotheses, the most studied theories focus on drug-target sensitivity, drug-transporters activity, neural network and intrinsic severity hypothesis [6]. Several evidences suggested that genes affecting the pharmacokinetic (absorption, distribution and metabolism) and pharmacodynamics processes, and the regulation of multidrug transporters, play a pivotal role in the resistance to antiepileptic drugs, explaining the diverse clinical response of patients with the same prognostic profile to the treatment with anticonvulsant (Figure 1). The genetic alterations may affect promoter, intronic and exonic regions of genes involved in different physiological pathways [8]. Recent efforts revealed genetic polymorphisms that may alter the sequence of the encoded protein, affect the turnover of RNA/protein and/or the efficiency of their transcription/translation. Although correlations with large gene expression profiles (transcriptomic and proteomics) are still missing, there are genetic variations that lead to drug resistance and emerging as potential candidates for refractory epilepsy, their screening may offer a therapeutic guide [9,10,11]. Appendix A summarizes genes whose alterations have been associated with no-responsive epilepsy to the treatment.

The completion of the human genome project highlighted the genetic variability among individuals, proving the importance of pharmacogenomics application in clinical practice for the administration of personalized therapies [12]. The task of pharmacogenomics is to recognize subgroups of patients with refractory epilepsy who share similar genetic profiles in order to prescribe therapeutic protocols that allow to have a good clinical outcome [13]. This approach represents undisputable support for clinicians in predicting patient response to medications when genetic variations are identified. The importance of the use of pharmacogenomics was considered also by the Clinical Pharmacogenetics Implementation Consortium (CPIC), who published guidelines on the use of CYP2C9 pharmacogenetic testing in phenytoin dosing in 2014 [14]. Moreover, also in AED package leaflets are provided indications about the side effects and efficacy of drug when genetic alterations (SNPs) occurring.

Concerning the mechanism of drug resistance, systems biology using a holistic scientific approach, can provide answers to complex questions related to the living systems like the brain: integrating data from genetic, genomic, neurophysiology or proteomic allows to identify pathways and network involved in the etio-pathogenesis of neurological diseases [15]. Moreover, focusing on the network interactions of molecular components, the systems biology enabling the discovery of new target drug resistance-associated [16].

In this review, we first describe the molecular mechanisms of refractory epilepsy, with special attention given to genetic alterations, and then introduce the use of a systems biology approach to decipher the pathophysiological complexity of epilepsy and lead to personalized diagnosis and treatments.

## 2. Heterogeneity in Pharmacoresistant Epilepsy

Heterogeneity of the epilepsy disorder affects patient management at various levels, leading in most of the cases to a delayed or incorrect diagnosis. As a consequence, the misdiagnosis of epilepsy prevents proper treatment and represents the main cause underlying refractory epilepsy. According to the International League Against Epilepsy guidelines, clinicians have to perform a differential analysis among 36 different forms of epilepsy based on seizure type, epilepsy type, epilepsy syndrome and etiology [17,18]. Four different epilepsy types are known: focal, generalized, combined focal and generalized, and unknown [19,20]. Focal seizures are distinguished into “focal aware seizures” and “focal impaired awareness seizures” depending on the maintenance or loss of consciousness during epileptic seizure. Furthermore, focal seizures are categorised according to their manifestation that may be “motor” (tonic, atonic, clonic epileptic spasms) or “non-motor” (cognitive, autonomic and emotional features). Tonic-clonic focal seizures can propagate to both hemispheres generating “focal bilateral tonic-clonic convulsions” [21]. Generalized seizures may have “motor” or “non-motor” (absence) features. Motor generalized seizures include clonic, tonic, tonic-clonic, myoclonic tonic-clonic, myoclonic atonic, atonic epileptic spasms, while absences are classified as typical, atypical, myoclonic, and absence with eyelid myoclonia [22]. This classification is based on the group of neuronal cells from which epilepsy originates without taking into consideration its genetic origin [23]. Other morbidity conditions that may look like epileptic disorders should be investigated, such as cognitive and behavioural delay, metabolic disturbance or head injury [19]. Correct classification of epilepsy represents the first step to achieve a more precise diagnosis and effective treatment. Genetic alterations not only contribute to etio-pathogenesis of epilepsy, but may also influence pharmacoresistance [24]. In the following sections, we will describe the main causes of pharmacoresistance and introduce a systems biology approach.

## 3. Drug Transporters in Pharmacoresistant Epilepsy

Understanding the mechanisms that lead the pathogenesis of resistance to antiepileptic drugs treatment is crucial to ensure better therapeutic strategies. In the last decades, the study of the role of the efflux transporters and their involvement in the antiepileptic drug resistance processes aroused growing interest. The best-known efflux transporters belong to the adenosine triphosphate-binding cassette (ABC) proteins superfamily, whose genes are highly conserved. These proteins act as ATP-driven membrane pumps that hydrolyze ATP, and leads to the carriage of a wide range of substrates against their concentration gradient across extra- and intra-cellular membranes. The spectrum of these transporters is very broad and includes amino acid, peptides, nutrients, sugars, pigments, metals and a large number of therapeutic drugs [25,26]. At first, the function and involvement of efflux transporters in drug resistance were examined in a wide range of tumor types [27]. Since multidrug transporter proteins are found to be constitutively expressed at the blood-brain barrier, it was hypothesized that their overexpression could be related to the antiepileptic drugs refractoriness, similar to chemotherapy-resistance [28]. Several lines of evidence indicated that the increased expression of these proteins in the blood-brain barrier enhance the extrusion of drugs from the target site, preventing the achievement of the drug minimum effective dose into the epileptic focus [28,29]. To date, seven subfamily of genes encoding efflux proteins are known: *ABCA (ABC1)*, *ABCB (MDR/TAP)*, *ABCC (MRP/CFTR)*, *ABCD (ALD)*, *ABCE (OABP), ABCF (GCN20)* and *ABCG* [25].

Initially, the correlation between pharmacoresistance and the overexpression of MDR1 mRNA levels in brain tissue of patients with refractory epilepsy was reported by Tishler and collaborators [30]. *MDR1* or *ABCB1* encodes the P-glycoprotein (P-gp) transporters, which allow the efflux of several antiepileptic drugs, including phenytoin, topiramate, carbamazepine, phenobarbital, lamotrigine, felbamate and gabapentin [1].

The first association-studies between pharmacological resistance in epilepsy and genetic alterations revealed a high frequency of the *MDR1* CC genotype [31], correlating the C3435T SNP with the increased expression of P-gp 170 [32]. Subsequently, several evidences confirmed that this polymorphism influenced the pharmacological response to antiepileptic drugs [33,34,35,36,37,38]. In addition, *ABCB1* homozygous polymorphisms, T1236C and G2677A, were the most frequent genotypes compared with the control group; a high frequency of the specific homozygous genotype G2677T/A in phenytoin-treated epileptic patients has been reported [36,37,38,39]. Current evidences revealed that individual polymorphisms contained in the *ABCB1* did not significantly influence the treatment outcome but rather the interaction between them could determine intractability in epilepsy [40,41,42,43]. Indeed, significant linkage disequilibrium was detected among C3435T, T1236C, and G2677T in *ABCB1*, proposing that the three loci jointly could affect drug responsiveness in epilepsy [36,38,41,44,45,46]. These results suggested that the assessment of the haplotypes in patients could be used as a prediction marker of drug resistance in epilepsy.

Although *ABCB1* was the candidate in association-study between pharmacoresistance and multidrug transporters with altered function, several investigations evaluated many other polymorphisms in different *ABC* genes. Interestingly, a correlation study proved the failure of the treatment with first-line antiepileptic drugs in patients bearing *ABCC2*-24T variant [47].

## 4. Cytochrome P450 Superfamily in Pharmacoresistant Epilepsy

A prevalent role in treatment failure or toxic effects of various pharmaceutical compounds can be ascribed to the variability in drug metabolism [48]. In approximately 90% of all administered drugs, biochemical pathway reactions are catalyzed by various cytochrome P450 enzymes (CYPs) [49,50,51]. So far, 57 genes encoding CYP enzymes have been cloned in the human genome and classified into different families (CYP1, CYP2, CYP3) and subfamilies (CYP1A, CYP2C), and further distinguished in several isoforms (CYP2C19, CYP2D6, CYP3A4) [52]. Alterations in genes encoding CYP450 result in a decreased or an increased enzyme activity that leads to excessive drug elimination before achieving a therapeutic action or an increase in drug concentration and permanence time in circulation, which may lead to undesirable effects [11]. These enzymes catalyze oxidative, peroxidative and reductive reactions, playing a pivotal role in drug interactions and interindividual variability in response to therapy. Many epidemiological studies correlated the highly polymorphic behavior of CYP450 genes with drug-resistant phenotypes. In particular, members of CYP1, CYP2, and CYP3 families are recognized as the most important hepatic enzymes involved in the metabolism of antiepileptic drugs. Among these, a great clinical relevance is blamed to genetic polymorphisms affecting *CYP2C9* and *CYP2C19*, *CYP2D6*, and *CYP3A4*. These polymorphisms have been studied in large populations, showing a great heterogeneity in the frequency of different alleles/genotypes. For example, the *CYP3A4*1B* and *CYP2C9*3* allelic variants, and *CYP2C9*3/*3* genotype are responsible of multiple drug-resistant phenotypes against antiepileptic drugs [49,53]. For additional details, the reader is referred to other studies [54,55,56].

## 5. Drug Target Sensitivity in Pharmacoresistant Epilepsy

The second emerging concept to explain pharmacoresistance contends that variations in the targets, voltage-gated ion channel properties or in genes encoding target molecules or receptors of antiepileptic drugs could reduce sensitivity to their ligands, affecting the response to the pharmacological treatment [57,58]. In addition, it is now widely accepted that the pathophysiological mechanisms promoting the occurrence and progression of seizures could be related to a balance alteration between the inhibitory GABAergic and the excitatory glutamatergic signaling pathways. Many of the currently available anticonvulsant drugs that control seizures act by increasing the Gamma Amino-Butyric Acid (GABA) signal, enhancing the flow of chloride ions, inhibiting neurotransmitter metabolism and reuptake, or modifying the voltage-gated Na^+^ sensitivity. Furthermore, antiepileptic drugs may regulate the activity of other voltage-gated channels, such as Ca^2+^ channels, interrupting the afferent and efferent pathways or reducing the glutamatergic signal [11,58,59].

### 5.1. The Voltage-Gated Na^+^ Channel

One of the primary targets of antiepileptic drugs used in both mono- and poly-therapy to control epileptic seizures is the voltage-gated Na^+^ channel. Numerous studies revealed, both in vitro and in vivo experimental models, an association between the reduction in response to antiepileptic drugs and an alteration of the Na^+^ channel structure and sensitivity [8,58,60].

The voltage-gated Na^+^ channel consists of one α subunit forming the transmembrane pore joined by two β subunits that show modulatory function. To date, nine α subunit isoforms (named Na_v_1.1–1.9; encoded by SCN1A-SCN11A) and four β subunits (β 1–4; encoded by SCN1B-SCN4B) are known. These channels exist in three different states: deactivated, active, and inactive. At resting membrane potential, the voltage-gated Na^+^ channels are in the deactivated state but rapidly switch to the active state upon depolarization, mediating the inward Na^+^ current. During prolonged depolarization, the channels undergo into the inactive state for some milliseconds, returning to the resting state only after hyperpolarization phase [61]. The transition among these functional states is therapeutically intriguing because antiepileptic drugs exhibit a different binding affinity level to the voltage-gated Na^+^ channels according to their conformation. Many first-line drugs in the treatment of epilepsy such as carbamazepine, valproic acid, phenytoin, lamotrigine, topiramate and oxcarbazepine exert their therapeutic action by reducing the Na^+^ channel-shooting time from the inactivation phase [62,63,64]. Thus, these drugs show low affinity for the resting channel state and high affinity when it is inactive, performing a more pronounced block of sodium inward during the increase neuronal excitatory state [65].

Different studies suggested that alterations in genes encoding Na^+^ channels produced a protein not responding to treatment with anticonvulsant drugs due to possible changes in the composition of ion channel subunits or to the expression of subunits non-sensitive to medicaments [66,67].

A genetic screening of the three main genes encoding α subunit Na^+^ channel, *SCN1A*, *SCN2A* and *SCN3A*, showed a positive correlation between *SCN1A* IVS5-91 G > A, *SCN2A* IVS7-32A > G, c.56 G → A and c.R19K polymorphisms and pharmacoresistance occurrence, highlighting the higher frequency of specific genotypes in patients resistant to the treatment with the Na^+^ channel-blocking antiepileptic drugs [68,69,70,71,72].

Moreover, alternative splicing is a crucial mechanism in the expression of Na^+^ channels that leads to different drugs-sensitivity or non-functional domains. Interestingly, a recent study focused on multidrug resistance process showed the involvement of three intronic *SCN1A* polymorphisms, rs6730344, rs6732655 and rs10167228, which could influence the final structure of the Na^+^-voltage channels through loss or creation of splice sites [73]. Although the α subunit is the most important functional component of sodium channel, experimental evidences showed that mutations in genes encoding the accessory β subunit might affect antiepileptic drug efficacy [65].

### 5.2. GABA_A_ Receptors

The second major class of genes involved in synaptic transmission linked to the unpredictability of efficacy of antiepileptic drugs encode the GABA_A_ receptor. GABA is the main neurotransmitter involved in the maintenance of the depressive state in mammalian central nervous system (CNS) and performs its inhibitory action on neuronal transmission through the binding to three receptor subtypes: Cl^−^-conducting ionotropic GABA_A_, GABA_C_, and metabotropic GABA_B_ receptors. GABA_A_ receptors are widely expressed in the CNS and, due to their fast response to GABA stimulation, represent the most relevant receptor subtypes in epilepsy therapy.

GABA_A_ receptor is structurally a pentameric ligand-gated channel consisting of eight subunits, α, β, γ, δ, ε, θ, π, ρ, each of which displays numerous subtypes. Gene sequencing studies allowed to identify six α, three β, three γ, one δ, ε, θ, π, and three ρ receptor subunit subtypes. Generally, GABA_A_ receptor is composed of two α and two β subunits, necessary for the correct formation of the ion channel, and a variable subunit among the others. The assembly of these different subunits and subtypes produces a large variety of GABA_A_ receptor combinations with heterogeneous structure, neuronal distribution, and sensitivity to ligands and drugs [74,75].

In the last decades, numerous experimental evidences suggested that the majority of GABA_A_ receptors in the CNS displays the 2α:2β:1γ stoichiometry, with a higher frequency of the α1β2γ2 subtype [76]. The binding of two GABA molecules to the interface between α and β subunits of GABA_A_ opens the chloride channel in the receptor, enhancing the influx of Cl^−^ and reducing the possibility of excitatory potential membrane trigger. In addition to the canonical GABA binding sites, a ligand-binding pocket in the GABA_A_ receptor is recognized by several drugs, many of which currently used in clinical practice for seizure control. Benzodiazepines and barbiturates are the main classes of medications used for the treatment of epilepsy both in mono- and poly-therapy that target the GABA_A_ receptor. In particular, benzodiazepines perform their anticonvulsant action through the binding at an allosteric site located between α and γ subunit of the GABA_A_ receptor. The interaction of these drugs at their binding pocket increases the receptor affinity for GABA neurotransmitter without inducing a direct inward of Cl^−^ through the ion channel. Therefore, benzodiazepines do not enhance the opening time of the hydrochloric channel but only the frequency, promoting the physiologic hyperpolarizing effect of GABA.

Based on these observations, it was proposed that alterations in genes encoding GABA_A_ receptor subunits, named *GABRx*, could be related to the lack of response to antiepileptic drugs [77]. The variability of the GABA_A_ receptor structure, render the understanding of the pharmacological resistance to treatment very complex and not immediate. Moreover, benzodiazepines display a different affinity for the subtype of the GABA_A_ receptor subunits: Lormetazepam shows a high affinity for the α1–3 and α5 GABA_A_ receptor subunits while carbamazepine and phenytoin interact prevalently with receptor composed of the α1 subunit; Topiramate enhance the Cl^−^ current inside the neuron affecting receptor made up to α1, α2 and α5 subunits [78,79,80]. Therefore, mutations or alterations in genes encoding α and γ subunits of GABA_A_ receptor induce aberrant formation of drugs-binding pocket, affecting clinical response to benzodiazepine-treatment. In support of this evidence, data describing a reduction of α1, α2, α5, β2/3 subunit levels concomitant to an enhanced expression of α4 subunits in rats with refractory epilepsy have been reported [77,81].

The importance of the γ_2_ subunit in benzodiazepine sensitivity was evaluated in *Xenopus* oocytes expressing wild type (WT) γ_2_-subunit and γ_2_ (R43Q) mutant-subunit in GABA_A_ receptors. Researchers observed that mutant receptors exhibited a reduced sensitivity to diazepam and flunitrazepam treatment compared to WT receptor [82]. In addition, investigation of some SNPs in *GABRA1* addressed the non-synonymous IVS11 + 15 A > G polymorphism as candidate for drug resistance, postulating its involvement in alternative splicing processes [83]. Further evidence revealed that the genotypic combinations of *GABRA1* rs6883877, *GABRA2* rs511310 and *GABRA3* rs4828696 influenced the response to drugs in non-sensitive epileptic patients, while the single gene variation did not affect therapy outcome [84].

### 5.3. Glutamate Receptors

Regulation of glutamate receptors plays a crucial role in the control of excitatory state in the CNS and therefore in the trigger of seizures. Despite the importance of this signaling pathway and the large amount of antiepileptic drugs available, few molecules targeting *N*-methyl-d-aspartate (NMDA) receptors are currently used in clinical practice.

NMDA receptors are heterotetrameric glutamate-gated ion channels composed of two GluN1 subunits, encoded by *GRIN1*, and two GluN2 or GluN3 subunits encoded by *GRIN2A-D* and *GRIN3A-B,* respectively [85]. The heterogeneity of subunit’s combination produces a wide variety of receptor subtypes with a different distribution, function, signaling, and pharmacological properties [86,87,88]. The interaction of glycine and glutamate to the specific binding pocket, located in the GluN1 and GluN2 subunits of NMDA receptor respectively, induces channel activation. Ligand-receptor association triggers the pore opening, which allows the entry of Ca^2+^ into the postsynaptic neuron concomitant with a voltage-dependent removal of magnesium (Mg^2+^) block strengthening membrane depolarization [89].

Whole-exome sequencing showed the presence of mutations in genes encoding NMDA receptor subunits in epileptic patients. Recent reports described *de novo* missense mutations, L812M and M817V, in *GRIN2A* coding for the GluN2A subunit in epileptic phenotypes. These variations resulted in increased glutamatergic activity, leading to an increased response to NMDA receptor agonists and to a reduction of negative modulator sensitivity, such as Mg^2+^ [90,91]. To further underline the potential of the genetic profiling analysis in the best treatment choice of epileptic patients, several polymorphisms that influenced the response to valproic acid treatment, were analyzed. Particularly, it has been reported that patients with the *GRIN2B*-200T > G polymorphism showed the same response to valproic acid but at a lower dose than non-carrier epileptics [92].

### 5.4. GABA_A_ and NMDA Receptors Trafficking

Alongside genetic mechanisms, several studies addressed resistance to antiepileptic drugs as an acquired and self-sustained phenomenon [2]. A good deal of evidence suggested that frequent epileptic discharges alter the receptor expression pattern on the plasma membrane of post-synaptic neurons modulating neuronal transmission and response to pharmacological treatment. It was widely demonstrated, in both in vitro and in vivo models, that the GABA_A_ receptor was rapidly internalized during recurrent seizures. This phenomenon led to a reduction in the inhibitory action of GABA, contributing to the maintenance of the epileptic status, and promoted the development of resistance to antiepileptic drugs targeting the GABA_A_ receptor [93,94,95,96]. Unsurprisingly, due to the heterogeneous nature of epileptic discharges, the self-perpetuating of seizures was also ascribed to modification in NMDA receptor activity and surface relocation. The role of NMDA receptors in maintaining epileptic status and in drug resistance has not been extensively studied, but some reports provide information on their activity and surface localization. Different evidences showed that the excessive neuronal stimulation enhanced trafficking of NMDA receptor to post-synaptic neuron surface, sustaining seizures activity [95,97]. Cho et al. demonstrated the fundamental role of magnesium in the trafficking patterns of GABA_A_ and NMDA receptors. Hippocampal neurons cultured in Mg^2+^-free medium showed a reduction in the GABA_A_ receptor expression concomitant with an increase NMDA receptor levels on the cell surface. Furthermore, since the action of the NMDA receptors were regulated by the presence of the Mg^2+^ block, the deprivation of this cation also caused an increase in the sensitivity of the glutamatergic receptors to their ligands, causing the trigger of continuous seizures [98]. Overall, these results show the limit of monotherapy and the benefit of polytherapy in perspective of a dynamic change in the receptor composition on the surface of post-synaptic neurons during seizures. The combined use of agents that enhance the GABAergic signaling pathway and inhibit the glutamatergic stimulation is essential for the successful outcome of the epilepsy pharmacological treatment [95,99].

## 6. Managing Drug Resistance Epilepsy in the Genomic Era

Epilepsy is a neurological disorder difficult to classify because of the different etiology, type of convulsions, epileptic syndromes, region and amplitude of the CNS involved. Due to the multifactorial etiology of epilepsy, the guidelines in epilepsy diagnosis require specific investigations based on clinical evaluation, EEG to determine seizure type, neuroimaging to identify structural abnormalities [100]. Given its heterogeneous nature, the choice of the therapeutic plan to control convulsions, minimizing the adverse effects, is under continuous exploration. The genetic counselling is considered one partner of this process and, pharmacogenomics studies, allowing the identification of genes that may play a role in drug response and toxicity, are fundamental to guide the treatment choice [101]. Up to now, the selection of the drug to treat epileptogenic episodes is based on empirical evaluations of the patient’s condition. Generally, a first-generation antiepileptic drug is administered as a first choice, the dose of which is established according to the severity of the epilepticus status: high doses to obtain immediate control of serious seizures; low doses, gradually increased, until the optimal dose is reached for the control of minor epileptic seizures. However, only a limited percentage of epileptic subjects became seizure-free after the administration of the first antiepileptic drug as monotherapy. Thus, to patients deemed pharmacologically treatment-resistant, a second or third antiepileptic drug is administered until switching to the polytherapy [17].

Development of high-resolution genomic technologies such as array-comparative genomic hybridization (aCGH) and next generation sequencing (NGS) allowed a breakthrough in the identification of copy number variations (CNVs) and genomic alterations linked to development of a wide range of disorders and therapeutic implications. These technologies can be used to fathom the entire human genome with unprecedented detail and draw a personalized profile of each patient [102]. In clinical practice, the application of genomic high-throughput technologies is now a routine to achieve the diagnosis but the use of the same genomic data to address the pharmacogenomics remains an attractive strategy.

Subjects with an epileptic phenotype caused by *SLC2A1* mutation and coding for Glucose Transporter Type 1 (GLUT-1) that prevents a correct glucose share to the CNS, represent a tangible example of the great potentiality of pharmacogenomics in epilepsy treatment. The identification of this mutation can obtain control of epileptic seizures by administering brain alternative energy sources through a ketogenic diet [103].

CNVs have been recently linked to pharmocoresistance in epileptic patients. Genome-wide characterization of CNVs detected recurrent microdelections and microduplications [104,105,106]. Particularly, several CNVs at loci 1q21.1, 15q11.2, 15q13.3, 16p11.2, 16p13.11, and 22q11.2 were associated to neurodevelopment disorders including autism spectrum, schizophrenia, attention problems, intellectual disability, dysmorphism, speech delay and epilepsy. Among them, the CNV at chr15q11.1–11.2, showed the most significant association to neurobehavioral disturbances and psychiatric problems (idiopathic generalized epilepsies [IGEs] or childhood absence epilepsy [CAE]) [104,105,106,107].

There is no doubt about the uselessness to evaluate genetic variations (SNPs or CNVs) without a clear correlation to the phenotype. However, extensive scientific evidence showed the crucial role of several genetic alterations on the mechanism of drug resistance and so take into account these correlations is now necessary.

The synergistic integration, annotation and analysis of genomic data led to the birth of a new field of functional biology, known as Systems Biology, which emphasizes and takes into account each molecular signature like a part of a systems network [108].

The systems biology approach is based on the combination of a large amount of data ranging from the analysis of simple cellular processes to complex networks. Indeed, the expression of single genes, the activity of codified proteins and the interaction of different pathways integrated in a wider network lead to obtain an overview of the complexity of biological phenomena [109].

By using a systems biology approach, in terms of our previous analysis that highlighted that different chromosomal aberrations—apparently unrelated to each other—may lead to the development of a pharmacoresistant phenotype. Specifically, it was an identified alteration in molecules acting in different pathways that contributed to the imbalance of the excitatory/inhibitory state and the onset of self-sustaining convulsions not responding to therapy. This evidence suggests that the integration of genomic data with a systems biology analysis helps to understand the role of crucial factors in the cross-talk of complex pathways, allowing us to predict a custom-made treatment based on health rather than disease.

Prospectively, given the high multifactoriality of drug resistance in epilepsy, the systems biology approach could represent a valid tool for pharmacological management. The integration of genomic, proteomic, and metabolomic data allows the identification of key factors that operate in complex networks, contributing to the development of multidrug resistance and, which could be targeted to overcome refractoriness. Thus, this approach represents a valuable resource to understand the individual susceptibility for disease or potential drug toxicity (Figure 2).

Ideally, the translation of systems biology in clinical practice to the management of epilepsy disease requires several changes in the medical field as the share of information and resources and the building of a network between all actors involved [110].

## 7. Conclusions

In recent decades, novel high-throughput technological platforms facilitate the prescription of therapies suited to the genomic characteristics and individual needs of the patient. The identification of individual genetic diversity allows the administration of the best drug or the correct medicament combination, reducing the risk of major adverse effects or inefficacy. To this regard, the introduction of the systems biology approach to establish the causative genotype-phenotype profile can enhance the exploitation of pharmacogenomic data [111]. The increasing availability of large-scale genetic data and the presence of more detailed clinical information and bioinformatic tools will certainly facilitate the spread of this new paradigm [112]. In oncology, biomarkers predicting response to anticancer treatment are providing the basis for individualized treatments. In neurology, we are still far behind but a more precise patient stratification will certainly contribute to advance the field of personalized medicine. The introduction of pharmacogenomics in refractory epilepsy, together with a systems biology approach is nowadays feasible and may allow a sensible reduction in morbidity and mortality in the near future.

## Figures and Tables

**Figure 1 ijms-20-03717-f001:**
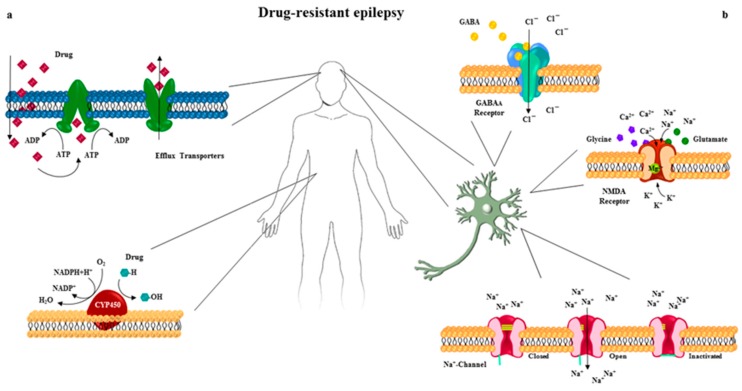
Schematic representation of the main pharmacokinetic (**a**) and pharmacodynamic processes (**b**) of antiepileptic drugs involved in refractory epilepsy. Mutations in genes encoding molecular targets could cause structural and/or functional alterations responsible for the lack of response to pharmacological treatment.

**Figure 2 ijms-20-03717-f002:**
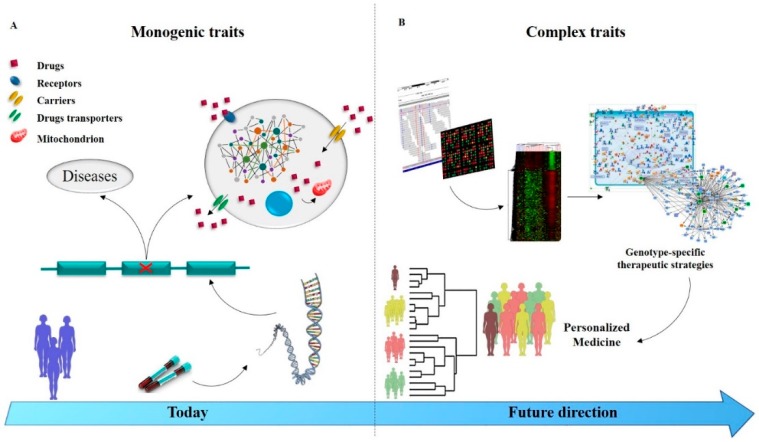
Illustrative representation of pharmacogenomics temporal evolution: from monogenic to complex genetic research analysis. (**A**) Genetic screening tests reveal variants correlated to the etiology of drug resistance in epilepsy. (**B**) Application of genetic technologies (NGS and CGH) integrated together with computational strategies allows the identification of casual disease networks and stratification of patients towards the personalized medicine.

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
