# Peer review of "A Systems Biology Approach for Personalized Medicine in Refractory Epilepsy"

_ijms, 2019, doi:10.3390/ijms20153717_

Round 1
Reviewer 1 Report
The manuscript titled “A systems biology approach for personalized medicine in refractory
epilepsy” discusses the current status of refractory epilepsy and potent contribution of genetic markers in treatment.
Authors have attempted to highlight an interesting aspect of epilepsy treatment; however, there are major limitations of this manuscript that need to be addressed.
1. After reading the manuscript, it leaves the reader to wonder if this manuscript is about pharmacological targets of refractive epilepsy or pharmacogenomics.
2. The manuscript is not well developed and appears to be an assortment of different concepts/facts. The concepts needs to be streamlined.
3. A Table should be included for PK and/or PD genes and polymorphism and epilepsy treatment outcomes.
4. There is no mention of CYP enzymes in this manuscript but this superfamily of enzyme is one of the major players in the refractory epilepsy.
5. Certain words/phrases are not conventionally used. For example, page 5 and 6, “deregulation”.
6. The conclusion of this manuscript is very generic . Needs to bring up what was discussed in the journal body.
Author Response
REVIEWER 1
“After reading the manuscript, it leaves the reader to wonder if this manuscript is about pharmacological targets of refractive epilepsy or pharmacogenomics.” “The manuscript is not well developed and appears to be an assortment of different concepts/facts. The concepts needs to be streamlined.”
Our article reviews the main causes underlying refractory epilepsy, focusing on the genetic alterations and proposing a systems biology approach that may lead to personalized treatment. In agreement with the referee’s comment, we have streamlined the manuscript and integrated it with additional paragraphs and references.
“A Table should be included for PK and/or PD genes and polymorphism and epilepsy treatment outcomes.”
The table has been revised.
“There is no mention of CYP enzymes in this manuscript but this superfamily of enzyme is one of the major players in refractory epilepsy.”
We have added a new paragraph related to CYP.
“Certain words/phrases are not conventionally used. For example, page 5 and 6, “deregulation.”
The manuscript has been fully revised.
“The conclusion of this manuscript is very generic. Needs to bring up what was discussed in the journal body.”
The conclusion has been revised.

Reviewer 2 Report
I find this article to be well constructed and written in its coverage of how a systems biology approach to refractory epilepsy may be developed and implemented. It can be a very positive contribution to the field.
One consideration that I would recommend to be included/added to this publication:
Epilepsy represents a heterogeneous clinical presentation with marked focus on the seizures that the patient experiences. I believe that this article would be significantly enhanced by adding a section that discusses the current understanding of the disease heterogeneity and the impact on diagnosis and stratification and whether/how this has been integrated into the evaluation of refractory epilepsy
Author Response
REVIEWER 2“Epilepsy represents a heterogeneous clinical presentation with marked focus on the seizures that the patient experiences. I believe that this article would be significantly enhanced by adding a section that discusses the current understanding of the disease heterogeneity and the impact on diagnosis and stratification and whether/how this has been integrated into the evaluation of refractory epilepsy.”
In agreement with the referee’s comment, we have included a new paragraph related to epilepsy heterogeneity.

Round 2
Reviewer 1 Report
The changes in the manuscript are satisfactory.